# The Effect of Microbial Diversity and Biomass on Microbial Respiration in Two Soils along the Soil Chronosequence

**DOI:** 10.3390/microorganisms10101920

**Published:** 2022-09-27

**Authors:** Jakub Vicena, Masoud M. Ardestani, Petr Baldrian, Jan Frouz

**Affiliations:** 1Institute for Environmental Studies, Charles University in Prague, Benátská 2, 12801 Prague, Czech Republic; 2Institute of Soil Biology and Biogeochemistry, Biology Centre, Czech Academy of Sciences, Na Sádkách 7, 37005 České Budějovice, Czech Republic; 3Laboratory of Environmental Microbiology, Institute of Microbiology of the Czech Academy of Sciences, Vídeňská 1083, 14220 Prague, Czech Republic

**Keywords:** carbon availability, decomposition of soil organic matter, fungal biomass, leaf litter, microbial biomass, microbial diversity

## Abstract

Microbial diversity plays an important role in the decomposition of soil organic matter. However, the pattern and drivers of the relationship between microbial diversity and decomposition remain unclear. In this study, we followed the decomposition of organic matter in soils where microbial diversity was experimentally manipulated. To produce a gradient of microbial diversity, we used soil samples at two sites of the same chronosequence after brown coal mining in Sokolov, Czech Republic. Soils were X-ray sterilized and inoculated by two densities of inoculum from both soils and planted with seeds of six local plant species. This created two soils each with four levels of microbial diversity characterized by next-generation sequencing. These eight soils were supplied, or not, by litter of the bushgrass *Calamagrostis*
*epigejos*, and microbial respiration was measured to assess the rate of decomposition. A strong positive correlation was found between microbial diversity and decomposition of organic matter per gram of carbon in soil, which suggests that microbial diversity supports decomposition if the microbial community is limited by available carbon. In contrast, microbial respiration per gram of soil negatively correlated with bacterial diversity and positively with fungal biomass, suggesting that in the absence of a carbon limitation, decomposition rate is controlled by the amount of fungal biomass. Soils with the addition of grass litter showed a priming effect in the initial stage of decomposition compared to the samples without the addition of litter. Thus, the relationship between microbial diversity and the rate of decomposition may be complex and context dependent.

## 1. Introduction

Soil organic matter (SOM) is an important source of stored carbon and is also important in terms of its global cycle [1,2,3,4,5]. In particular, newly formed soils that are not carbon-saturated can sequester a large amount of carbon [6,7]. SOM is irreplaceable for the nutrient cycle; has buffering and detoxifying ability; is an environment for the accumulation, retention, and infiltration of water; and is indispensable in terms of substance transformation [8,9]. 

Decomposition is a key process in terms of carbon cycle and other elements in terrestrial ecosystems [10]. The decomposition of SOM is estimated to release up to 60 Gt of carbon per year on a global scale [11]. Therefore, the decomposition mechanisms and factors that affect decomposition are keys to better understanding and predicting global climate change. In the context of climate change caused by industrial development, transport, and agriculture, future changes in the rate of decomposition are expected [12].

Microorganisms in soil, namely bacteria, fungi, and archaea, are responsible for decomposition [13]. Due to their widespread presence, bacteria represent key soil organisms responsible for decomposition [14,15]. Fungi are highly efficient in the decomposition of resistant compounds, namely lignocellulose [16]. Moreover, fungi can compete with bacteria by producing various allelochemicals [17]. By competing for resources, modification of soil environment, and various other interactions, fungi substantially shape bacterial niches in soil [18]. Next-generation approaches will substantially increase our knowledge about microbial diversity and community composition in various ecosystems [19,20,21,22,23,24]. Many studies have proposed that increasing microbial diversity will increase the ability of the microbial community to deliver some functions, such as litter decomposition [25,26]. However, the relationship between microbial diversity and the rate of individual functions in soil is still poorly understood [27,28,29]. Some studies have shown an increase in decomposition with increasing microbial diversity [29,30,31], whereas others showed that remarkably different microbial communities may have the same effect on the rate of decomposition in the same condition [32].

Microbial diversity and the composition of the microbial community are controlled by soil pH, humidity, temperature, and the diversity of vegetation cover [33,34,35]. These factors may also affect the decomposition process directly, which complicates detangling the effect of microbial diversity on decomposition. In addition, contradictory results of various effects of microbial diversity on decomposition suggest that these effects are context dependent. In agreement with the stress gradient hypothesis, one may expect that positive interactions are more important when resources are limited [36]. Microbial diversity has been often reported to be positively correlated with microbial respiration, which may be assumed as a proxy for organic matter decomposition [37]. However, the effect of changes in microbial diversity vs. biomass changes has been seldom addressed. It is possible that changes in biomass will have larger impact on microbial respiration than changes in microbial diversity.

In this study, we compare two post-mining soils (young and old) with natural differences in their microbial diversity. These soils were sterilized and backwards inoculated by the inoculation of both soils with different levels of dilution. This created a gradient of microbial diversity in two soils, which allowed us to obtain comprehensive information about the impact of microbial community diversity on the decomposition of organic matter. We hypothesized that microbial diversity supports the decomposition of SOM when resources are depleted and less decomposable; but, when resources are likely to be less depleted, the biomass of major decomposers will be more important.

## 2. Materials and Methods

### 2.1. Sampling, Preparation, and Experimental Setup

This experiment used soil that was produced in the soil inoculation experiments described in Frouz et al. [38]. The soils were collected from two dumps after brown coal mining of different ages in the Sokolov region of northwestern Czech Republic. The deposited material consisted of alkaline clay stones, which subsequently decomposed into smaller particles and amorphous clay. During the soil formation process on this chronosequence, the pH gradually decreased, phosphorus became more available, and the amount of stored carbon and nitrogen gradually increased [39]. 

The intention was to select soils of different ages and chemical composition, which were more likely to indicate significant differences in terms of microbial community diversity as well as microbial biomass. This could result in different decomposition speeds. The first soil type was marked as E (early succession) and the age was approximately 10 years. Clay stone was decomposed into particles <2 mm and the vegetation cover was low. Soil pH was determined to be 8.75 [38]. The soil from the second site was 50 years old and designated as L (late succession). The influence of roots, soil fauna, and vegetation at this site for several decades formed a soil horizon with a thickness of 8–10 cm. The soil pH was less alkaline than younger soil and it was 7.16 [38].

Sampling was performed at both locations at the same time. Five samples were taken from a depth of 5–8 cm. The distance between individual samples was approximately 50 m. The samples from each site were pooled into two composite samples. Approximately 1 kg of substrate was stored at 4 °C. Leftover substrate was placed in four resealable plastic bags and sterilized by 40 kGy of γ-radiation. Sterilized substrates were inoculated by a suspension of unsterilized soil to obtain 10^−2^ and 10^−7^ dilutions [38,40]. In this case, dilution was the proportion of soil that was used to form the inoculum relative to the sterilized soil. The suspension was subjected to sonication and filtered through a 40 μm filter. The suspension volume added to each iteration was always the same. The difference was the use of different amounts of non-sterile substrate to form a suspension for each treatment.

This procedure resulted in eight soil combinations: LLH, LLL, LEH, LEL, ELH, ELL, EEH, and EEL (Appendix A). We kept this coding to be consistent with the earlier study in which the same soils and similar experimental setup was used [38]. The first letter indicates the type of sterile substrate used (E—younger succession stage, L—older succession). The second letter indicates the soil from which the suspension was formed, through which the sterile soil was subsequently inoculated (E—younger soil, L—older soil). The third letter indicates which dilution was used for each treatment. The 10^−2^ dilution is marked as H (high diversity, less diluted) and 10^−7^ dilution as L (low diversity, more diluted). We expected lower presence of microorganisms at greater dilution [41]. In the present study, we used all eight soil combinations (treatments) from Frouz et al. [38] except LEL soil, which was not sufficient to establish this treatment.

Glass jars with tight-closing lids were used to measure the microbial respiration of the samples with a volume of 150 mL. Prior to the start of the experiment, these vessels were sterilized. Soils from individual treatments were weighed in a flow box to reduce the likelihood of contamination by microorganisms from the environment. Forty grams of soil was weighed into each vessel in 2–6 replicates depending on how much material was available. Two groups of samples were created: samples with and without the addition of 0.5 g of bushgrass *Calamagrostis epigejos* L. (Poaceae) leaf litter (samples with “-Cal” refer to the treatments with the addition of litter hereafter). The *C. epigejos* leaves were stored in 3 × 3 cm polyester bags and sterilized using the same method as soil samples. The bags were placed in the soil containers so that they were covered with substrate. Subsequently, 5 mL of deionized water was added to the soil. The experiment lasted for 1100 days. Approximately in the middle (570 days) and before the end (1000 days) of the whole experiment, 2 mL of deionized water was added to each sample.

### 2.2. Measurements

#### 2.2.1. Microbial Respiration

Microbial respiration was used as a proxy for the rate of decomposition. The static respirometry method was used to measure the microbial respiration. To do so, microcosm was hermetically closed and carbon dioxide release by microbial respiration was absorbed in 0.5 M NaOH, placed in the microcosm in a small beaker. Initially, we used 6 mL NaOH, and later on, as respiration rate decreased, this volume was decreased by up to 3 mL. The incubation time of sodium hydroxide in the microcosm was 1 week. The amount of CO_2_ released was determined by titration using HCl [42].

Data from Frouz et al. [38] were used to determine respiration with respect to available carbon. The amount of available carbon was measured in the monitored soils. In the case of the late succession stage, this value was 7.8% C (carbon), but it was 2.4% C for the soil in the early succession stage [38]. The amount of carbon contained in the added *C*. *epigejos* litter was set at 0.2 g (i.e., 40% carbon in the leaves). Subsequently, the amount of available carbon was calculated based on these values in individual microcosms. For late soil without the addition of litter, it was 3.12 g C, and for the same soil with the addition of litter, it was 3.32 g C. Early soil without litter addition contained 0.96 g C and, with the addition of litter, it had 1.16 g C [38]. Respiration with respect to available carbon was then determined by dividing the respiration measured by the respective values for available carbon. 

#### 2.2.2. Microbial Community Characteristics

To quantify the microbial biomass, microbial biomass carbon was measured by fumigation extraction. Ergosterol content was used as a proxy of fungal biomass. In addition, for the bacterial community, we studied bacterial richness using next-generation sequencing as described below in details.

Using the fumigation extraction method, microbial biomass carbon was measured directly, which is counted as released carbon from microbial cells after soil fumigation in the chloroform vapor [43]. The soil sample was divided into two parts. Soluble carbonaceous substances were extracted from the first part, which was affected by chloroform vapors (fumigation) and compared to non-fumigated samples [40,44,45].

Ergosterol is a substance found in the cell membranes of fungi. It is produced only by living microorganisms and decomposes very quickly after their death. The method of determination is based on the extraction of ergosterol from the soil and subsequent analysis on a liquid chromatograph with spectrophotometric detection. The measurement itself was performed on a DIONEX ICS-5000 instrument (Thermo Scientific, Poway, CA, USA). Ergosterol was detected by a UV detector at 282 nm. The concentration of ergosterol was determined automatically based on peak area using calibration standard curves.

Before the experiment, the bacterial community composition was evaluated by the 454 pyrosequencing method. The results of this effort and detailed methodology are described in Frouz et al. [38]. Briefly, genomic DNA was extracted from soil samples as described by Sagová-Marečková et al. [46] and then amplified by PCR. The first amplification step used the eubacterial primers eub530F and eub1100aR to amplify the V4-V6 hypervariable regions of the bacterial 16S rDNA gene. Each soil sample was purified using the Wizard SV Gel and PCR Clean-Up System (Promega, Madison, WI, USA). The purified DNA concentration was determined using ND1000 (Nano-Drop, Wilmington, DE, USA). In the second amplification step, fusion primers were tailored for tag-encoded 454 Titanium pyrosequencing: different barcode sequences were added at the 5ʹ end of the forward primer separated by a trinucleotide spacer. The Titanium A adaptor was also used (Roche, Basel, Switzerland). PCR products were purified with the MinElute PCR Purification Kit (Qiagen, Hilden, Germany) and quantified with the ND1000 (NanoDrop) and the Quant-iT Picogreen dsDNA Assay Kit (Invitrogen, Carlsbad, CA, USA). Purified amplicons were used for the subsequent emulsion PCR (emPCR Kit Lib-L, Roche, Basel, Switzerland), the products of which were sequenced on a GS Junior platform (Roche) in accordance with the manufacturer’s instructions. Sequences were processed with the QIIME 1.6.0 software package. Quality filtering steps were performed to trim off barcodes and primers from the raw sequences and to remove sequences that were <200 nt long, that had homopolymers longer than 6 nt, and that had a quality score <25. Denoising was performed as described by Reeder and Knight [47]. The QIIME’s implementation of OTUPipe script [48,49] was applied for chimera checking and OTU picking. The 16S and 18S Microbial BLAST databases were used as reference databases for bacterial and fungal chimeric sequence detection, respectively. Resulting chimera-free reads were clustered into OTUs based on their sequence similarity at 97%. Representative sequences of each OTU were aligned using MUSCLE [50] and used for taxonomy assignment. The BLAST database was used to taxonomically classify the bacterial and fungal sequences. Alpha-rarefaction and alpha-diversity analyses were performed with QIIME 1.6.0, and beta-diversity analyses were performed with weighted UniFrac [51]. The jack-knifed beta diversity tree was used for clustering of individual soils before and after plant growth. Pyrosequencing data were uploaded in the Metagenomics RAST IDs: 4612656.3.

### 2.3. Data Analysis

The obtained data were organized in Microsoft Office Excel. For the evaluation of homogeneous groups and significant differences between treatments, we used one-way analysis of variance (ANOVA) and repeated measurements ANOVA followed by Fischer LSD post hoc test. To evaluate mutual correlations, the analysis of Pearson correlation coefficients was used for all parameters except for litter addition, soil age, and inoculum, where Spearman Rank correlation was used. All statistical computations were completed in the program Statistica (TIBCO^®^, version 13.3; Software for Statistical Computation; USA). To compare microbial respiration between treatments, we used cumulative respiration during the whole experiment, in addition to that, we paid specific attention to the initial stages of decomposition when respiration sharply decreased, and then on later stages when respiration stabilized. The idea behind this division is that initial decomposition may be affected by substrate manipulation and consumption of easily available C released by sterilization, drying, and rewetting, while the later stage gives some estimation of respiration after this initial flush of easily available C has been consumed.

## 3. Results

The highest operational taxonomic unit (OTU) richness was determined for ELH soils and was close to the value in the EEL soil (Table 1). The OTU richness was determined in the LLH sample, and a slightly higher number in the LLL sample. Therefore, we can state that the OTU richness was higher for the substrate-based treatments in early soil (Table 1).

High initial respiration occurred in the samples with the addition of litter during the initial phase of decomposition, but respiration sharply decreased with time (Figure 1A). Later, the decline was slowed down. The rate of decomposition was gradually decreased for the rest of the experiment, with a few exceptions. Ascending trends followed by a further decline was observed in 570 days and 1000 days, when the samples were hydrated. In soils without the addition of litter, the initial decrease in respiration was not noticeable, but we observed a slight increase (Figure 1B). 

Figure 2A shows a significant difference in the average decomposition rate of the samples with the addition of litter during and after the first seven (initial decomposition phase) and the following seven measurements (later decomposition phase). As already indicated, the samples with the addition of litter had a higher average respiration in the initial phase of decomposition. Samples without the addition of litter followed an opposite trend with higher average respiration at a later stage of decomposition (Figure 2B). 

The microbial respiration measurements showed significant differences in decomposition. This trend applied to soils without the addition of *C. epigejos* litter vs. soils in which litter was added (Figure 3). Average respiration corresponded well to the values observed in Figure 4, which shows the average microbial biomass carbon. The highest average respiration was 11.45 C-CO_2_ h^−1^ microcosm^−1^ and the highest value for microbial biomass carbon per gram dry soil was 1599 μg C g^−1^ in the sample LLL-Cal. Lower values among late soils were measured for the LLH-Cal sample (7.59 μg C-CO_2_ h^−1^ microcosm^−1^ and 774 μg C g^−1^). Samples based on substrate E were lower and the differences were not as significant as with the substrate L samples (Figure 3 and Figure 4). A similar trend was observed for samples without the addition of litter. The highest average respiration and microbial biomass carbon were also measured in the LLL sample (8.46 μg C-CO_2_ h^−1^ microcosm^−1^ and 1133 μg C g^−1^). 

The average respiration relative to available carbon differed significantly (Figure 5) from the results for respiration that was not related to the amount of carbon (Figure 3). The respiration values for the substrate E-based treatments in this case were significantly higher than the values for treatments of late soil (Figure 5). Higher respiration values compared to the available carbon was observed in the samples based on soil substrate E. An example was soil with the highest respiration per gram of available carbon, EEH-Cal (5.25 C-CO_2_ h^−1^ microcosm^−1^ g C_available_^−1^), or EEL-Cal soil (4.87 C-CO_2_ h^−1^ microcosm^−1^ g C_available_^−1^). 

A significantly higher ergosterol content was evident in the treatments of late soil (Figure 6). The ergosterol content was highest in the LLL-Cal sample with the addition of litter (0.87 ppm). For samples without the addition of litter, the highest value was observed for the LLL sample (0.76 ppm). In soils without the addition of litter, the trend for treatments of L substrate was similar to soils with litter addition. However, values recorded for early soil were at an unmeasurable level (Figure 6).

The analysis of correlation coefficients between the parameters of individual soils showed that microbial respiration in the whole microcosm correlated with microbial biomass and ergosterol content and increased with the addition of litter (Table 2). In contrast, bacterial diversity expressed by the OTU richness correlated negatively with microbial respiration. It may be related to a negative correlation between the bacterial OTU richness and microbial biomass and ergosterol content. However, if we look at respiration converted to available carbon in the microcosm, the OTU richness and addition of litter correlated positively with respiration, whereas the amount of carbon and soil age negatively correlated (Table 2).

## 4. Discussion

### 4.1. Relationship between Diversity, Respiration, and Biomass of Microbial Community

In agreement with our original hypotheses, our results show that the relationship between microbial diversity and the rate of decomposition is contextual. Microbial respiration per unit of soil carbon had a positive correlation with microbial diversity, whereas respiration per microcosm correlated negatively with microbial diversity and positively with microbial biomass. This suggests that microbial diversity increases decomposition in the presence of low organic matter availability, whereas decomposition is related to fungal biomass in high organic matter availability and negatively relates to microbial diversity. This conceptuality may explain why some other authors found that microbial diversity increases decomposition [29,30,31], whereas others found a negative correlation between diversity and decomposition [23]. A positive correlation between decomposition with higher microbial biomass has been presented in other studies [52,53,54]. 

Ergosterol, which is a good proxy for living fungal biomass [55,56,57,58], increased with available carbon and litter addition. This agrees with the general understanding that fungi are important decomposers of plant litter [13,59]. We also found that bacterial diversity decreased with an increasing amount of available carbon, and microbial biomass was increased. A decrease in bacterial diversity with increasing microbial biomass was also observed by Luo and Gu [24]. Bacterial diversity had a negative relationship with litter addition, which may be explained by competition with fungi as proposed by other authors [60,61,62]. This is also supported by the negative correlation between ergosterol and microbial diversity.

In summary, this relationship between microbial diversity and decomposition seems to be context dependent and depends on available organic matter coming from the soil and added litter. Similar context-dependency was also observed by Maron et al. [29], who explained this phenomenon by nutrient availability. This may not contradict our results when we consider the amount of nutrients available per unit of carbon. We observed that less available organic matter led to an increase in decomposition due to higher microbial diversity. When litter is added, decomposition of the whole system becomes negatively related to bacterial diversity and driven by fungal biomass and total microbial biomass. In other words, when resources are scarce, microbial diversity supports the decomposition process and, when fresh litter is more plentiful, fungal biomass is increased. In contrast, bacterial diversity is decreased (likely due to competition) and decomposition increased simultaneously. Interestingly, a similar phenomenon was observed in the plant community. In a situation with low resource availability, diversity increases plant production, whereas under high resource availability, plant diversity decreases and production increases [63,64]. This suggests that diversity is important when resources are limited. When resources are plentiful, diversity may be decreased, and function depends rather on the overall number of organisms. This may be a more general ecological phenomenon. 

In our study, we used dilution to create gradient of microbial diversity. Although this happened, it was not exactly as expected. In some treatments, more diluted material led to higher bacterial richness than less diluted ones. This is probably caused by the fact that more complex interactions between inoculum and inoculated substate may happen as discussed in details in Frouz et al. [38].

### 4.2. Influence of Litter and Water Addition on Respiration

Low carbon availability is well-known to limit the energy flow passing through the soil microbial community and, consequently, the rate of mineralization of SOM [65]. The effect of increasing the proportion of fresh SOM results in a sharp increase in microbial activity associated with a higher rate of decomposition [65]. This phenomenon, also referred to as the “priming effect”, has been the subject of several studies [66,67,68], but its mechanisms are still unclear [69,70]. In agreement with our hypothesis, we observed an increase in respiration, which may be attributed, in part, to the priming effect. With the addition of litter, a significantly higher average decomposition rate was seen in the initial phase than in the later phase. In samples in which no litter was present, we observed higher average respiration at a later stage of decomposition. The possible answer is an absence of litter at the same time as available nutrients. This would be consistent with the theory of delayed microbial growth biomass compared to samples in which nutrients are more available [71,72,73].

Respiration in the microbial community is strongly influenced by water availability [74,75]. In our experiment, we observed declining respiration in soil without litter addition and in soil with added litter. After the addition of water, the decomposition rate significantly increased, which reached a peak in the sample without the addition of litter in approximately 40 days. For samples with litter, we observed the peak about a month later. These results led us to the idea that the intensity and length of the response for decomposition based on the addition or loss of water could vary depending on the biomass of the microbial community and on its functional (division according to energy sources) and species composition (microbial diversity). This could correspond to very different decomposition reactions to the addition or lack of water, which were observed with different soils in different parts of the world [75,76,77,78]. Namely, the prevalence of bacteria in soil without litter addition, which likely react quickly upon the addition of water, and the prevalence of fungi in soils with added litter, which likely react more slowly upon the addition of litter [79,80].

## 5. Conclusions

The decisive factor in the dependence of decomposition on microbial community diversity is a limitation of available carbon. If carbon is not limiting, the decomposition rate decreases with greater diversity and a quantitative effect of microbial biomass prevails. In the case of carbon limitation, the dependence of decomposition on microbial diversity manifests, and the influence of microbial biomass does not play a significant role. Decomposition is faster for samples with a larger amount of fungal biomass. The amount of fungal biomass is dependent on the entry of litter and the age of the soil. The relationship between microbial and fungal biomass is strongly dependent on the given conditions. In soils with the addition of litter, the priming effect caused in the initial phase was confirmed by the respiratory curves. The influence of the addition of water on the increase in microbial activity is faster in bacteria-dominated soil than in soil with higher fungal biomass.

## Figures and Tables

**Figure 1 microorganisms-10-01920-f001:**
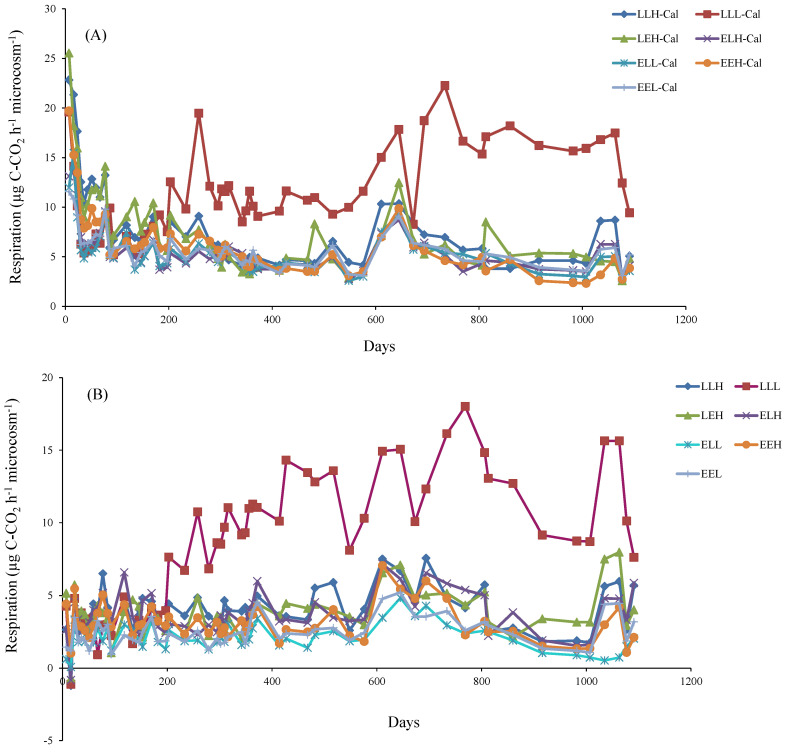
The average soil microbial respiration with the addition of *Calamagrostis epigejos* leaf litter (**A**) or without the addition of leaf litter (**B**) throughout the experiment (μg C-CO_2_ h^−1^ microcosm^−1^).

**Figure 2 microorganisms-10-01920-f002:**
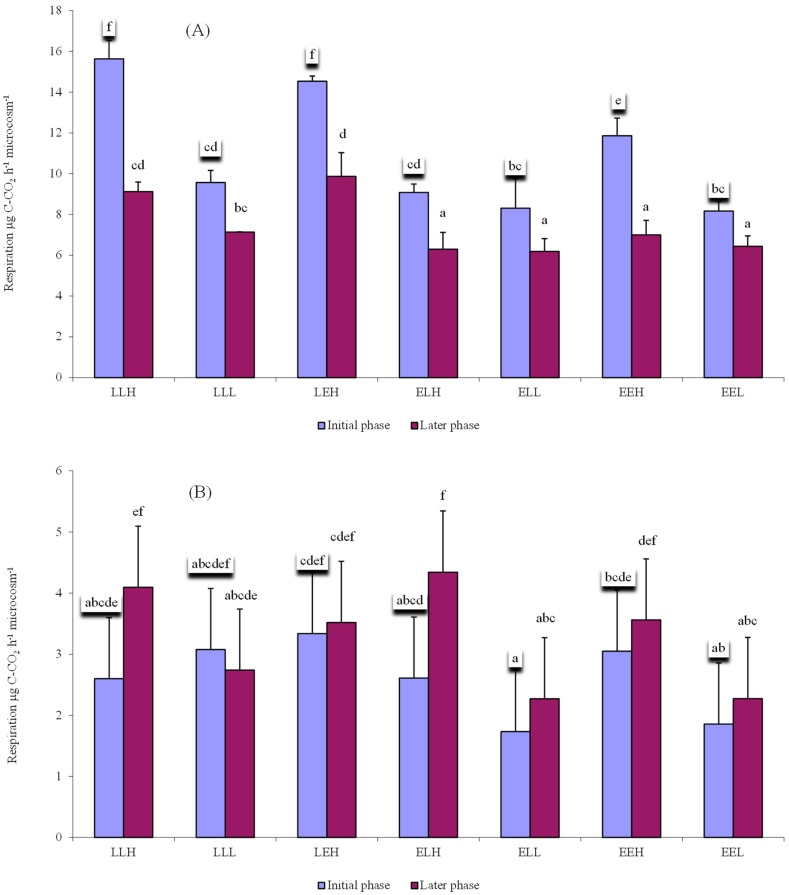
Average respiration (μg C-CO_2_ h^−1^ microcosm^−1^) of individual treatments with the addition of litter (**A**) or without the addition of litter (**B**) in the first 51 days (initial phase) and the subsequent 83 days (later phase). Error bars are standard deviation. Results are based on repeated measurement ANOVA. If ANOVA for the soil type was significant, the results of the post hoc test (Fischer LSD, *p* < 0.05) are given. Homogeneous statistical groups are shown by the same letters.

**Figure 3 microorganisms-10-01920-f003:**
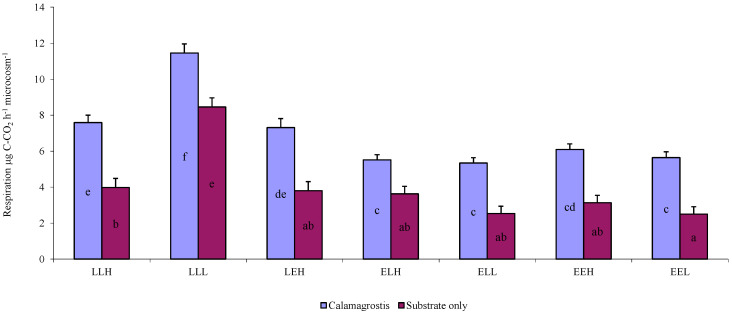
Average respiration of individual treatments with the addition of litter *Calamagrostis epigejos* (Calamagrostis) or without the addition of litter (Substrate only) for the whole experiment time (μg C-CO_2_ h^−1^ microcosm^−1^). Error bars are standard deviation. Results are based on repeated measurement ANOVA. If the ANOVA was significant for the soil type, the results of the post hoc test are given (Fischer LSD, *p* < 0.05). Homogeneous statistical groups are indicated by the same letters.

**Figure 4 microorganisms-10-01920-f004:**
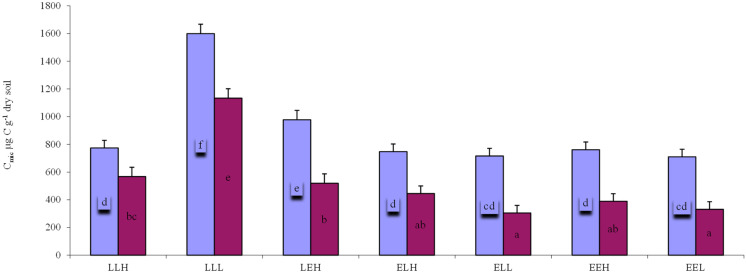
Average microbial biomass carbon (C_mic_) per gram of dry soil at individual treatments (μg C g^−1^) with the addition of *Calamagrostis epigejos* litter (Calamagrostis) or without the addition of litter (Substrate only). Error bars are standard deviation. Results are based on one-way ANOVA. If the ANOVA was significant for the soil type, the post hoc results are given (Fischer LSD, *p* < 0.05). Homogeneous statistical groups are indicated by the same letters.

**Figure 5 microorganisms-10-01920-f005:**
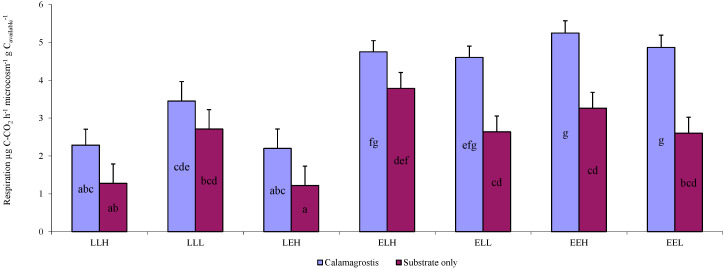
Average respiration of individual treatments with the addition of litter *Calamagrostis epigejos* (Calamagrostis) or without the addition of litter (Substrate only) for the whole experiment time related to the amount of available carbon (C-CO_2_ h^−1^ microcosm^−1^ g C_available_^−1^). Error bars are standard deviation. Results are based on repeated measurement ANOVA. If ANOVA was significant for soil type, the results of the post hoc test are given (Fischer LSD, *p* < 0.05). Homogeneous statistical groups are indicated by the same letters.

**Figure 6 microorganisms-10-01920-f006:**
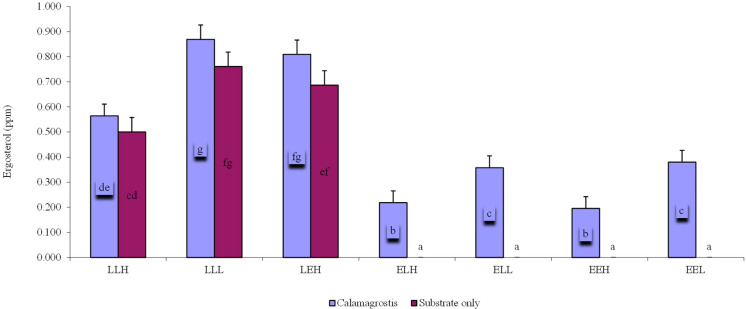
Average ergosterol content (ppm) for individual treatments with the addition of litter *Calamagrostis epigejos* (Calamagrostis) or without the addition of litter (Substrate only). Error bars are standard deviation. Results are based on one-way ANOVA. If ANOVA was significant for soil type, the results of the post hoc test are given (Fischer LSD, *p* < 0.05). Homogeneous statistical groups are indicated by the same letters.

**Table 1 microorganisms-10-01920-t001:** Operational taxonomic unit (OTU) richness found in individual soil samples, amount of available carbon in substrate for individual treatments, and soil pH taken from Frouz et al. [38].

	LLH	LLL	LEH	ELH	ELL	EEH	EEL
OTU richness	69	79	87	127	86	111	125
Available carbon	3.12	3.12	3.12	0.96	0.96	0.96	0.96
Soil pH	7.16	7.16	7.16	8.75	8.75	8.75	8.75

The first letter for each treatment indicates the type of sterile substrate used (E—younger succession stage, L—older succession). The second letter indicates the soil from which the suspension was formed, through which the sterile soil was subsequently inoculated (E—younger soil, L—older soil). The third letter indicates which dilution was used for each treatment. Dilution of 10^−2^ is marked as H and dilution of 10^−7^ as L. Available carbon is in grams per microcosm or per gram of soil.

**Table 2 microorganisms-10-01920-t002:** Regression coefficients for the parameters of individual soils.

	Microbial Respiration	Microbial Respiration per Gram C	Ergosterol	Microbial Biomass	Available C per Microcosm	Number of OTU
Ergosterol	0.76	-	-	-	-	-
Microbial biomass	0.97	-	0.78	-	-	-
Available C per microcosm	0.60	−0.55	0.88	0.59	-	-
OTU richness	−0.39	0.51	−0.63	−0.36	−0.76	-
Addition of litter	0.64	0.52	0.38	0.57	-	-
Area age	0.54	−0.61	0.85	0.54	1.00	−0.77
Substrate age from which inoculum was obtained	0.63	-	0.63	0.64	0.65	−0.54

The addition of bushgrass *Calamagrostis epigejos* L. (Poaceae) litter was coded as 0 (without litter addition) or 1 (with litter addition) and the age of the area as 1 (younger area) or 2 (older area). Only significant correlation coefficients are displayed (*p* < 0.05).

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
