# Peer review of "The Effect of Microbial Diversity and Biomass on Microbial Respiration in Two Soils along the Soil Chronosequence"

_microorganisms, 2022, doi:10.3390/microorganisms10101920_

Round 1

Reviewer 1 Report

Relationship between microbial diversity and its function attracts much attention, which has long been debated in the previous literature. This study is valuable in revealing the relationship between microbial diversity and organic matter decomposition by creating diversity gradient with two soils. I just have some minor concerns which will be listed as follow.

1.      The title for the manuscript is not good, which does not well reflect the content and aims of the present study. Please think about it and present a more appropriate one.

2.      In the abstract, the authors stated “In this study, we followed the decomposition of litter in soils---”. Actually, this study mainly focused on the decomposition of organic matter.

3.      What does “final process” mean at the end of the abstract? This is a bit confusing!

4.      The last sentence in Page one is an unclear presentation, which needs to be reworded.

5.      I was confused by “---we can anticipate future changes in the rate of decomposition in the future [12]”.

6.      To my understanding, hypothesis comes from logical reasoning. How can you state that “--- in litter decomposition, the biomass of major decomposers will be more important”. I don’t see any discussion for this statement in the introduction section.

7.      It is hard to understand that “We observed that, when organic matter was less available and taken from soil organic matter, microbial diversity increased decomposition.” in the third paragraph of 4.1. Can you rephrase it?

Author Response

Comments for the manuscript Microorganisms-1910307

Reviewer 1:

Relationship between microbial diversity and its function attracts much attention, which has long been debated in the previous literature. This study is valuable in revealing the relationship between microbial diversity and organic matter decomposition by creating diversity gradient with two soils. I just have some minor concerns which will be listed as follow.

Answer: Thanks for the positive comments of the reviewer 1 for our manuscript. We considered your concerns and made small changes in the text.

  1. The title for the manuscript is not good, which does not well reflect the content and aims of the present study. Please think about it and present a more appropriate one.

Answer: We changed title to: The Effect of Microbial Diversity and Biomass on Microbial Respiration in Two Soils Along the Soil Chronosequence

  1. In the abstract, the authors stated “In this study, we followed the decomposition of litter in soils---”. Actually, this study mainly focused on the decomposition of organic matter.

Answer: We changed this.

  1. What does “final process” mean at the end of the abstract? This is a bit confusing!

Answer: We changed it.

  1. The last sentence in Page one is an unclear presentation, which needs to be reworded.

Answer: We removed it.

  1. I was confused by “---we can anticipate future changes in the rate of decomposition in the future [12]”.

Answer: We rephrased it.

  1. To my understanding, hypothesis comes from logical reasoning. How can you state that “--- in litter decomposition, the biomass of major decomposers will be more important”. I don’t see any discussion for this statement in the introduction section.

Answer: Microbial diversity has been often reported to be positively correlated with microbial respiration which may be assumed as a proxy for organic matter decomposition. However, the effect of changes in microbial diversity vs. biomass changes has been seldom addressed. It is possible that changes in biomass will have larger impact on microbial respiration than changes in microbial diversity. We explained that in Introduction section.

  1. It is hard to understand that “We observed that, when organic matter was less available and taken from soil organic matter, microbial diversity increased decomposition.” in the third paragraph of 4.1. Can you rephrase it?

Answer: We rephrased it.

Reviewer 2 Report

Article title:

Microbial Community Diversity Influences the Decomposition of Organic Matter in Two Soils Along the Soil Development Gradient"

 I recommend the publication of the article because scientific experimentation which allowed us to obtain comprehensive information about the impact of microbial community diversity on the decomposition of organic matter.  

The work done is certainly of international interest and the format applied is certainly suitable for a research article. The work is original, of particular interest, and can certainly stimulate research on this topic. The length of the article is appropriate for the journal and the graphs and tables are clear and easy to understand. The conclusion summarizes the aims of the work and future prospects.

Author Response

Comments for the manuscript Microorganisms-1910307

Reviewer 2:

Article title:

“Microbial Community Diversity Influences the Decomposition of Organic Matter in Two Soils Along the Soil Development Gradient"

I recommend the publication of the article because scientific experimentation which allowed us to obtain comprehensive information about the impact of microbial community diversity on the decomposition of organic matter. 

The work done is certainly of international interest and the format applied is certainly suitable for a research article. The work is original, of particular interest, and can certainly stimulate research on this topic. The length of the article is appropriate for the journal and the graphs and tables are clear and easy to understand. The conclusion summarizes the aims of the work and future prospects.

Answer: Thanks for the positive comments of the reviewer 2 for our manuscript.

Reviewer 3 Report

In the manuscript by Vicena et al., the authors describe an experiment that investigates how soil microbial communities from sites with two different soil ages and differing levels of bacterial richness influence decomposition. Bacterial richness was positively correlated with soil respiration per gram of carbon, but negatively correlated with soil respiration per gram of soil. This suggests that bacterial diversity may influence decomposition when carbon is not limiting. Because soil respiration per gram of soil also positively correlated with ergosterol concentration, the authors conclude that decomposition in soils where carbon may be more limiting is controlled by the abundance of fungi in the soil. Overall, I think the experimental design is well done. I have several comments about the way the data are analyzed and reported, especially given that the conclusions stem from data reported in a different study. Thus, I think the study can be improved by clarifying more information about the microbial richness data and by running the ANOVA models in a different fashion.

Major comments:

1.     Because the microbial community data are part of a different study (Frouz et al.), which I do not have access to, it is difficult for me to know where the OTU richness numbers are coming from. I think they are bacterial community data, even though the authors use microbial consistently throughout the manuscript. Sometimes this can include other groups, such as archaea, microbial eukaryotes, and even fungi. Please include more information in the methods about the 454 sequencing that was used, especially the primer set that was used to PCR-amplify the microbial DNA. Please also be consistent throughout the manuscript as to which microbial community was studied and manipulated for the low (10-7 dilution) and high (10-2 dilution) diversity treatments.

2.     Table 1 shows the OTU richness numbers that the low and high diversity treatments are based on. Since the final letter in the sample name indicates either low (L) or high (H) diversity, the OTU numbers presented in Table 1 don’t reflect these treatments. For example the LLL sample has 79 OTUs, while the LLH sample has 69 OTUs. I would expect higher OTU numbers in the samples with the “H” at the end. Is there another measure of diversity (Shannon, Simpson, evenness, etc.) in Frouz et al. that is being used to define the low and high diversity treatments? If not, then it’s not accurate to call these treatments low and high diversity based on the richness numbers reported in Table 1. If anything, the treatments are low and high soil dilutions; perhaps with changing microbial biomass, which the authors did measure and did not report the numbers of. I suggest that the authors provide better support of the low and high diversity with their soil dilutions or use a different term for this treatment that includes sufficient data to also support the new terminology (such as microbial biomass differences between the dilutions).

3.     Overall, I disagree with how the authors ran their ANOVA models. Instead of using two- or three-way ANOVAs where litter addition is included in the model, the authors run one-way ANOVAs on the litter absent microcosms and then on the litter present microcosms and then attempt to make comparisons between the litter treatments. This should be done with an ANOVA that has multiple factors. Also, the correlations that are used for the main conclusions are not sufficiently described in the methods, so it’s difficult to know what the numbers are in Table 2. I am detailing my comments about statistics by manuscript section or figure/table number, as I think this will help the authors make revisions:

a.     Section 2.3: More information is needed for the correlation coefficients in this section. Were these Spearman Rank or Pearson correlations? How were they calculated? Table 2 has “regression” listed, so were these coefficients reported from regressions or correlations? Please clarify.

b.     Figures 2, 3, and 5: Since these are average respiration values, I don’t think a repeated measures ANOVA is necessary. The element of time has been removed from the data, especially in figures 3 and 5 where the respiration values were averaged over the entire experiment.

c.     Figure 2: How are the two phases defined? This was not detailed in the methods, but needs to be. The results section states “with the addition of litter during and after the first seven (initial decomposition phase) and the following seven measurements (later decomposition phase).” But, there are far more than seven measurements for this study. Please explain in both the methods and results.

d.     Figures 2, 3, 4, 5, and 6: First, I suggest combining Figures 3 and 5 since these the conclusions that the authors draw are based on comparisons between respiration per gram of soil and respiration per gram of C. This will allow the reader to better see the differences in the patterns that the authors describe. Second, I really think the authors need to re-consider the one-way ANOVAs used here. I suggest at least a two-way ANOVA where soil microbial treatment is one factor and litter addition is another factor. It seems that the authors have separated the two litter treatments and performed one-way ANOVAs on either the plus litter samples or minus litter samples. By running multiple tests, this increases Type I error. In addition, by including both factors, it will allow a test for interactions so the authors can see if the plus litter samples respond to microbial manipulation differently than the minus litter samples.

e.     Table 2: The inclusion of litter addition, soil age, and inoculum for these correlations is problematic. These are categorical variables, not continuous, with two categories each. These are the treatments that are tested with the ANOVA models and do not need to be included here.

f.      Finally, a large portion of the conclusions is based on three correlations in Table 2. If the ANOVA models are run in a different fashion, they can be used to support the same conclusions to a better extent. For example, in the absence of the litter, ergosterol was undetected in the carbon-limited early soils. This supports the conclusion that, in the absence of carbon limitation, fungi become a dominant decomposer. I suggest that the authors use the ANOVAs to support their conclusions more.

Minor comments (it is difficult to do this without line numbers, but I’ll do the best I can):

Introduction, paragraph 2:  Just from an editorial standpoint, there are redundant terms in the final sentence (decomposition twice and future twice).

Introduction, paragraph 3: Fungi are more than what is described here. Estimates are that fungi are significant contributors to decomposition in soils and they have a larger role than what is described here. I suggest expanding on what is currently written. See, for example, de Boer et al., 2005 https://doi.org/10.1016/j.femsre.2004.11.005 and articles that cite this paper.

Introduction, paragraph 3, references 8-23: It seems like these papers all deal with decomposition to some extent. Do you mean to include decomposition in the sentence here? Otherwise, this sentence about next-generation approaches and microbial communities is very broad and should include a host of review articles.

Introduction, paragraph 5: For the hypothesis sentence, can you please clarify what is different for litter decomposition? Does it make it so that resources are no longer limited?

Section 2.1, paragraph 2: What aspect of the microbial community? Please clarify if you mean composition, diversity, etc.

Section 2.1, paragraph 2: I suggest softening the language here, and throughout the manuscript. I would use “This could result…” instead of “This would result…” for example.

Section 2.1, paragraph 4: The use of visa versa here is confusing. I suggest defining “H” as “high diversity.”

Section 2.1, paragraph 4: The final sentence is really confusing. Please re-phrase to indicate that the previous experiment used all soil for this treatment.

Section 2.2, paragraph 2: Here and in other instances, the world “measurement” is used when I think the authors mean “experiment.” For example, the first sentence of this paragraph should read “The experiment lasted for 1100 days.”

Section 2.2, paragraph 3: This reads more like results to me than methods. There is nothing here about how these percent carbon values were measured or how carbon was estimated in the bushgrass litter. This needs to be clarified a bit. Were all carbon data taken from the other paper? Even the bushgrass? How did you get from 7.8% to 3.12 g C? Was it based on the mass of soil in the 150 mL? Please include equations.

Section 2.2, paragraphs 4 and 5: Please be clear with what the fumigation extraction method and ergosterol method are measuring. Was total microbial biomass measured with the fumigation extraction or just microbial biomass carbon? Both Table 2 and the Discussion mention total microbial biomass, but it isn’t clear from the methods how this was measured.

Results section, paragraph 2: Again here, “…rest of the experiment…” not “…rest of the measurement…”

Results section, paragraph 2: When was the slight increase in respiration noticed? Was it for all samples? It isn’t clear at what time point this was seen from the graph except for the LLL sample.

Results section, paragraph 4: I’m confused how the respiration measurements showed significantly different decomposition. Wasn’t respiration the measurement used to determine decomposition? Please rephrase.

Results section, paragraph 4: There is an explanation of how E substrates were not as significant as L substrates. This is where doing 2-way or 3-way ANOVAs could help. Maybe there is a difference between E and L source soils with the varying inocula that would manifest as a significant interaction. The ANOVA would include source soil and inocula soil as two factors and maybe litter as a third factor.

Results section, paragraph 5: You need a statistical test to tell if the two respiration calculations differed significantly, which was not done. I think you mean that by standardizing the respiration values to available carbon, a different pattern emerged.

Results section, paragraph 5: What is “-Cal”? Please define or remove.

Results section, paragraph 6: Typically in scientific writing we use past tense. Please change “is” to “was” here and check throughout the manuscript.

Section 4.1, paragraph 1: I think the authors mean “fungal biomass” and not “microbial biomass” here. It was ergosterol that correlated positively with respiration per gram of soil. But, there are several mentions of microbial biomass in the Discussion and in Table 2. Was this measured with the fumigation extraction method? Please describe in the methods if so.

Section 4.1, paragraph 1: Instead of stating that “This means…” change to “This suggests…”. The results here are based on correlations and correlation does not equal causation. Soften the language here and throughout the Discussion please.

Section 4.2, paragraph 2: What is the control? This was not explained in the methods. Please explain or remove.

Author Response

Comments for the manuscript Microorganisms-1910307

Reviewer 3:

In the manuscript by Vicena et al., the authors describe an experiment that investigates how soil microbial communities from sites with two different soil ages and differing levels of bacterial richness influence decomposition. Bacterial richness was positively correlated with soil respiration per gram of carbon, but negatively correlated with soil respiration per gram of soil. This suggests that bacterial diversity may influence decomposition when carbon is not limiting. Because soil respiration per gram of soil also positively correlated with ergosterol concentration, the authors conclude that decomposition in soils where carbon may be more limiting is controlled by the abundance of fungi in the soil. Overall, I think the experimental design is well done. I have several comments about the way the data are analyzed and reported, especially given that the conclusions stem from data reported in a different study. Thus, I think the study can be improved by clarifying more information about the microbial richness data and by running the ANOVA models in a different fashion.

Answer: Thanks for the comments of the reviewer 3 for our manuscript. We tried to make small changes in the text and clarify information about microbial analysis.

Major comments:

  1. Because the microbial community data are part of a different study (Frouz et al.), which I do not have access to, it is difficult for me to know where the OTU richness numbers are coming from. I think they are bacterial community data, even though the authors use microbial consistently throughout the manuscript. Sometimes this can include other groups, such as archaea, microbial eukaryotes, and even fungi. Please include more information in the methods about the 454 sequencing that was used, especially the primer set that was used to PCR-amplify the microbial DNA. Please also be consistent throughout the manuscript as to which microbial community was studied and manipulated for the low (10-7 dilution) and high (10-2 dilution) diversity treatments.

Answer: We added some sentences about the sequencing in the Method section.

  1. Table 1 shows the OTU richness numbers that the low and high diversity treatments are based on. Since the final letter in the sample name indicates either low (L) or high (H) diversity, the OTU numbers presented in Table 1 don’t reflect these treatments. For example the LLL sample has 79 OTUs, while the LLH sample has 69 OTUs. I would expect higher OTU numbers in the samples with the “H” at the end. Is there another measure of diversity (Shannon, Simpson, evenness, etc.) in Frouz et al. that is being used to define the low and high diversity treatments? If not, then it’s not accurate to call these treatments low and high diversity based on the richness numbers reported in Table 1. If anything, the treatments are low and high soil dilutions; perhaps with changing microbial biomass, which the authors did measure and did not report the numbers of. I suggest that the authors provide better support of the low and high diversity with their soil dilutions or use a different term for this treatment that includes sufficient data to also support the new terminology (such as microbial biomass differences between the dilutions).

Answer: We agree with that however we wanted to use the same coding as Frouz et al., 2016 SBB paper; we mentioned that in the Method section. We also added some comments in the Discussion pointing on why less diluted inoculum may not mean higher microbial diversity and refer to Frouz et al., 2016 paper where this issue is explained in more details.

  1. Overall, I disagree with how the authors ran their ANOVA models. Instead of using two- or three-way ANOVAs where litter addition is included in the model, the authors run one-way ANOVAs on the litter absent microcosms and then on the litter present microcosms and then attempt to make comparisons between the litter treatments. This should be done with an ANOVA that has multiple factors. Also, the correlations that are used for the main conclusions are not sufficiently described in the methods, so it’s difficult to know what the numbers are in Table 2. I am detailing my comments about statistics by manuscript section or figure/table number, as I think this will help the authors make revisions:

Answer: We agree that using multiple ANOVA is conceptually correct. In our case, we should use soil, litter addition, inoculum, and dilution as a factor. This will lead to 4-way ANOVA. However, 4-way ANOVA will consume too much degree of freedom and reduce sensitivity of the test, and increasing risk of second type error. Considering this, we decided to use one-way ANOVA as more robust solution which we believe brings similar answer to our research questions. 

  1. Section 2.3: More information is needed for the correlation coefficients in this section. Were these Spearman Rank or Pearson correlations? How were they calculated? Table 2 has “regression” listed, so were these coefficients reported from regressions or correlations? Please clarify.

Answer: We used Pearson correlation coefficient except of litter addition, soil age, and inoculum where Spearman Rank correlation was used, using default setting of Statistica 6, we added this in the Method section.

  1. Figures 2, 3, and 5: Since these are average respiration values, I don’t think a repeated measures ANOVA is necessary. The element of time has been removed from the data, especially in figures 3 and 5 where the respiration values were averaged over the entire experiment.

Answer: We explained it above.

  1. Figure 2: How are the two phases defined? This was not detailed in the methods, but needs to be. The results section states “with the addition of litter during and after the first seven (initial decomposition phase) and the following seven measurements (later decomposition phase).” But, there are far more than seven measurements for this study. Please explain in both the methods and results.

Answer: To compare microbial respiration between treatments, we used cumulative respiration during whole experiment, in addition to that we paid specific attention on initial stages of decomposition when respiration sharply decreased, and then on latter stages when respiration stabilized. The idea behind this division is that initial decomposition may be affected by substrate manipulation and consumption of easily available C released by sterilization, drying and rewetting, while latter stage gives some estimate of respiration after this initial flush of easily available C has been consumed. We explained this in data analysis part of the Method section.

  1. Figures 2, 3, 4, 5, and 6: First, I suggest combining Figures 3 and 5 since these the conclusions that the authors draw are based on comparisons between respiration per gram of soil and respiration per gram of C. This will allow the reader to better see the differences in the patterns that the authors describe. Second, I really think the authors need to re-consider the one-way ANOVAs used here. I suggest at least a two-way ANOVA where soil microbial treatment is one factor and litter addition is another factor. It seems that the authors have separated the two litter treatments and performed one-way ANOVAs on either the plus litter samples or minus litter samples. By running multiple tests, this increases Type I error. In addition, by including both factors, it will allow a test for interactions so the authors can see if the plus litter samples respond to microbial manipulation differently than the minus litter samples.

Answer: We explained it above.

  1. Table 2: The inclusion of litter addition, soil age, and inoculum for these correlations is problematic. These are categorical variables, not continuous, with two categories each. These are the treatments that are tested with the ANOVA models and do not need to be included here.

Answer: We used Spearman Rank correlation which is generally assumed to be suited well for non-normal and even categorial variables, we mentioned that in the Method section.

  1. Finally, a large portion of the conclusions is based on three correlations in Table 2. If the ANOVA models are run in a different fashion, they can be used to support the same conclusions to a better extent. For example, in the absence of the litter, ergosterol was undetected in the carbon-limited early soils. This supports the conclusion that, in the absence of carbon limitation, fungi become a dominant decomposer. I suggest that the authors use the ANOVAs to support their conclusions more.

Answer: We explained it above.

Minor comments (it is difficult to do this without line numbers, but I’ll do the best I can):

Introduction, paragraph 2:  Just from an editorial standpoint, there are redundant terms in the final sentence (decomposition twice and future twice).

Answer: We rephrased this sentence.

Introduction, paragraph 3: Fungi are more than what is described here. Estimates are that fungi are significant contributors to decomposition in soils and they have a larger role than what is described here. I suggest expanding on what is currently written. See, for example, de Boer et al., 2005 https://doi.org/10.1016/j.femsre.2004.11.005 and articles that cite this paper.

Answer: We completely agree and we expanded this part of the Introduction highlighting that fungi in large extend shape bacterial niches in soil. We used the suggested reference.

Introduction, paragraph 3, references 18-23: It seems like these papers all deal with decomposition to some extent. Do you mean to include decomposition in the sentence here? Otherwise, this sentence about next-generation approaches and microbial communities is very broad and should include a host of review articles.

Answer: As a matter of fact, only first half of the paragraph deals with decomposition, while second part deals with microbial communities and fungi bacterial interaction. We added decomposition and communities in appropriate sentences to separate these two sections better.

Introduction, paragraph 5: For the hypothesis sentence, can you please clarify what is different for litter decomposition? Does it make it so that resources are no longer limited?

Answer: We agree that we really did not measure resource limitation and we reformulated the hypothesis accordingly.

Section 2.1, paragraph 2: What aspect of the microbial community? Please clarify if you mean composition, diversity, etc.

Answer: We explicitly mentioned microbial diversity and biomass here.

Section 2.1, paragraph 2: I suggest softening the language here, and throughout the manuscript. I would use “This could result…” instead of “This would result…” for example.

Answer: We changed this.

Section 2.1, paragraph 4: The use of visa versa here is confusing. I suggest defining “H” as “high diversity.”

Answer: We rephrased this sentence.

Section 2.1, paragraph 4: The final sentence is really confusing. Please re-phrase to indicate that the previous experiment used all soil for this treatment.

Answer: We rephrased this sentence.

Section 2.2, paragraph 2: Here and in other instances, the world “measurement” is used when I think the authors mean “experiment.” For example, the first sentence of this paragraph should read “The experiment lasted for 1100 days.”

Answer: We changed this in the text.

Section 2.2, paragraph 3: This reads more like results to me than methods. There is nothing here about how these percent carbon values were measured or how carbon was estimated in the bushgrass litter. This needs to be clarified a bit. Were all carbon data taken from the other paper? Even the bushgrass? How did you get from 7.8% to 3.12 g C? Was it based on the mass of soil in the 150 mL? Please include equations.

Answer: It refers to the work of Frouz et al. 2016 who have these estimations and numbers in their study.

Section 2.2, paragraphs 4 and 5: Please be clear with what the fumigation extraction method and ergosterol method are measuring. Was total microbial biomass measured with the fumigation extraction or just microbial biomass carbon? Both Table 2 and the Discussion mention total microbial biomass, but it isn’t clear from the methods how this was measured.

Answer: We made small changes in the method to describe it better.

Results section, paragraph 2: Again here, “…rest of the experiment…” not “…rest of the measurement…”

Answer: We changed this.

Results section, paragraph 2: When was the slight increase in respiration noticed? Was it for all samples? It isn’t clear at what time point this was seen from the graph except for the LLL sample.

Answer: We mentioned here LLL as an example for all samples.

Results section, paragraph 4: I’m confused how the respiration measurements showed significantly different decomposition. Wasn’t respiration the measurement used to determine decomposition? Please rephrase.

Answer: We made small changes in the Method section to explain this better.

Results section, paragraph 4: There is an explanation of how E substrates were not as significant as L substrates. This is where doing 2-way or 3-way ANOVAs could help. Maybe there is a difference between E and L source soils with the varying inocula that would manifest as a significant interaction. The ANOVA would include source soil and inocula soil as two factors and maybe litter as a third factor.

Answer: We explained it above.

Results section, paragraph 5: You need a statistical test to tell if the two respiration calculations differed significantly, which was not done. I think you mean that by standardizing the respiration values to available carbon, a different pattern emerged.

Answer: We explained it above.

Results section, paragraph 5: What is “-Cal”? Please define or remove.

Answer: We added words to clarify this.

Results section, paragraph 6: Typically in scientific writing we use past tense. Please change “is” to “was” here and check throughout the manuscript.

Answer: We changed this and checked throughout the manuscript.

Section 4.1, paragraph 1: I think the authors mean “fungal biomass” and not “microbial biomass” here. It was ergosterol that correlated positively with respiration per gram of soil. But, there are several mentions of microbial biomass in the Discussion and in Table 2. Was this measured with the fumigation extraction method? Please describe in the methods if so.

Answer: We changed this. We added few words to the Methods to describe this better.

Section 4.1, paragraph 1: Instead of stating that “This means…” change to “This suggests…”. The results here are based on correlations and correlation does not equal causation. Soften the language here and throughout the Discussion please.

Answer: We changed this.

Section 4.2, paragraph 2: What is the control? This was not explained in the methods. Please explain or remove.

Answer: We rephrased it.